# Roles and Effect of Digital Technology on Young Children's STEM Education: A Scoping Review of Empirical Studies

Xinyun Hu [1,*,†], Yuan Fang [2,†]  and Yutong Liang [1]

1 Department of Early Childhood Education, The Education University of Hong Kong, Hong Kong SAR 999077, China; cycliang@eduhk.hk
2 Department of Health and Physical Education, The Education University of Hong Kong, Hong Kong SAR 999077, China; lunajoef@gmail.com
* Correspondence: xinyunhu@eduhk.hk
† These authors contributed equally to this work.

**Abstract:** Digital technology is increasingly used in STEM education for young children aged 0–8 years. An extensive literature search was conducted using seven databases to systematically investigate the effect of digital technology on young children's STEM education. Twenty-two eligible articles published from 2010 to 2021 were identified. Results showed that robotics, programming, and multimedia were used to support young children's STEM education. Digital technology plays different roles in the process of STEM education. Outcomes also showed that digital technology positively affected young children's STEM education in terms of STEM knowledge or skill acquisition and learning engagement. This was regardless of gender but relevant to age and the learning condition. Participating children and teachers reported high acceptance and satisfaction with the included programs. However, many difficulties, challenges and criticisms were revealed by the extracted data, including how digital technology is used in young children's STEM education, the nature of young children, the requirements placed upon educators, and different types of adult–child interactions. We also look at the limitations of the study design within included studies and provide recommendations accordingly.

**Keywords:** digital technology; early childhood education; effectiveness; learning engagement; STEM

## 1. Introduction

Across the globe, Science, Technology, Engineering, and Mathematics (STEM) education is increasingly necessary in early childhood education. Indeed, STEM education is seen as essential by educators and policymakers, as well as in business and industry, to meet the rapidly evolving needs of a technological society [1]. Interest in STEM education began in the West and has spread to many developing countries [2–4]. This indicates a desire to boost STEM education across various learners, including young ones [5]. The core of STEM education necessitates that individual learners develop an in-depth understanding of STEM-related concepts and skills while learning so that they can solve problems in real-world scenarios [2,6]. STEM education is effective in enhancing cognitive skills and boosting the robustness of behavioral competencies [7,8]. However, current STEM education is commonly studied and emphasized in children in higher grades of primary schools or middle schools [7,9], which triggered the research questions of to what degree STEM education is also effective in early childhood education (0–8-year-old children) [10] and how to reinforce the efficiency to impact on learning.

Moreover, the gender difference in science achievement is documented as significant among students of secondary and postsecondary schools [11–13]. The stereotype threat helps to harm women's confidence and interest in these traditionally masculine areas [14,15]. Whereas their interest in science may boost female participation in STEM-related fields in early education or by their family, males typically attribute their interest in STEM to

their internal curiosity [16,17]. Therefore, a theory-based scientific curriculum without a stereotyped environment may increase the motivation to learn science in young children, regardless of gender [17,18]. However, the female disparity in science achievement may be adversely affected by their experience in the earliest grade in school, which is related to policy [19]. To combat this, early exposure to STEM curriculum and programming may reduce gender-based stereotypes regarding STEM career orientation in young children [20].

STEM education for young children is a comprehensive and interdisciplinary learning journey that emphasizes the integration of science, technology, engineering, and mathematics. Many developed countries or regions, including Australia [21], England [22], New Zealand [23], and Hong Kong [24], recognize the importance of providing children with a holistic and interconnected learning experience that closely relates to their daily lives. In line with this, the Hong Kong Education Kindergarten Curriculum Guide (KECG) encourages teachers to incorporate real-life themes into multidisciplinary learning rather than focusing solely on individual disciplines [24]. Moreover, children are expected to apply their knowledge and skills to solve practical problems in their everyday lives, such as "*using mathematical concepts to solve practical problems in everyday life*" [24] (p. 39). Consequently, STEM education for young children is considered an "*effort to combine some or all of the four disciplines of science, technology, engineering and mathematics into one class, unit or lesson that is based on connections between the subjects and real-world problems*" [25].

STEM education seeks to engage children with knowledge and skills in the real world [26–28]. Yelland [29] states, "*STEM education in the early years provides a context for designing active learning ecologies that connect with children's natural curiosity about their world. It systematically engages children in authentic investigations, using critical and creative thinking to build knowledge, acquire skills, and cultivate confident dispositions for learning*" (p. 240). Early engagement in STEM education provides equity for girls' future STEM education and careers, and no substantial gender gap in science achievement was found among young children in their early childhood [19]. Indeed, this may further promote children's STEM-related skills and learning engagement, such as math skills, and social–emotional development [30]. However, STEM education in early childhood is still a developing field. As such, there may not be comprehensive disciplinary coverage in early integrated STEM education [30,31], and engagement with different disciplines may be unbalanced [31–35]. For instance, engineering-based learning often prioritizes science-related subjects over mathematics education [34]. Researchers [32,33] also highlight the undervaluing of engineering-related thinking and skills in classroom practice in early STEM education. Furthermore, it is important to note that early STEM education sometimes focuses on single subjects rather than interdisciplinary, integrated learning, such as mathematics-focused STEM-related learning [36] or science-driven STEM activity design [37].

Digital technologies are frequently used to provide STEM knowledge and enhance learning engagement. These technologies include robotics, programming, 3D technology, games or apps, touchscreen devices, computers, smartphones, television, etc. [38–42]. The assistance of digital technologies for young children is supposed to take similar roles as previous studies reported (i.e., learning from technology, learning with technology, and learning through technology).

Learning from technology occurs when technology is a tool to store and deliver knowledge. As such, learning can be limited to the content used by the technology [43]. In contrast, learning with technology occurs when learners use technology to access, organize, interpret, and analyze information. This allows learners to obtain new knowledge, regardless of the technology's limits [43,44]. Learning through technology occurs when users use technology as a platform to generate new and valuable technology from the original technology, which may create a technology-enhanced learning environment for teachers and learners [45].

Empirical data showing digital technologies' effect on children's learning of integrated STEM is limited. According to the published reviews, Slavin et al. [46] indicate that technologies-integrated teaching and cooperative learning show positive outcomes in

science achievement measures among a few small matched studies conducted in elementary schools. They imply that technologies can support teachers and enhance instruction in a single subject, such as science learning. Jung and Won [40] investigate the current strategy of robotics education in pre-kindergarteners to 5th grade primary children. They found that constructivist and constructionist frameworks mainly were used both in robotics curriculum design and the evaluation of learning engagement. Herodotou [47] focuses on digital tablets. She found them to positively affect kindergarten children's literacy development, mathematics, science, problem-solving, and self-efficacy. Wan et al. [42] summarize the positive attitudes and perceptions of STEM among parents of kindergarten children whilst documenting the various practical concerns of teachers regarding the lack of resources and self-efficacy.

*Objectives*

The current scoping review was conducted to summarize the key findings from the empirical studies conducted before the COVID-19 pandemic (2020–2021). The research questions included the following:

(1) What types of digital technology have been adopted to assist young children's STEM education?
(2) What role does digital technology play in assisting young children's learning of STEM knowledge/skills?
(3) What level is digital technology effective in assisting young children's STEM education in terms of acquiring knowledge/skills and learning engagement?
(4) What factors have been identified as related to the effect of digital technology on young children's STEM education?

## 2. Materials and Methods

### 2.1. Search Strategy

The academic articles were identified by searching the electronic databases. These include Web of Science (Clarivate PLC, London, UK; including SSCI and A&HI) and EBSCOhost (EBSCO Industries, Inc., Vincent, AL, USA; including ERIC, Education Full text, Education Research Complete, APA PsycArticles, and APA PsycINFO)

These cover the publication periods of 1905–2021 and 1975–2021, respectively. The search period was 2010–2021. The Boolean operator was used in the search strategy conducted with "OR" and/or "AND" to link search terms. The asterisk "*" was used as a wildcard symbol appended at the end of the terms to search for variations of those terms. The completed search process is listed in Table 1.

**Table 1.** Search strategy.

| No. | Searched Items in the Topic/Abstract |
|-----|--------------------------------------|
| 1 | "STEM" |
| 2 | "technology" OR "computer" OR "tablet" OR "mobile" OR "mobile phone" OR "smartphone" OR "internet" OR "TV" OR "television" OR "app*" OR "digital toy" OR "iPad" OR "website" OR "robotic" OR "robot" OR "computer technology" OR "digital media" OR "ICT" OR "computer programming" OR "3D printing" OR "3D printer" OR "virtual reality" OR "VR" OR "argument reality" OR "AR" OR "360-degree video" OR "littleBits" OR "Internet of Toys" OR "IoToys" |
| 3 | "early childhood" OR "early years" OR "early childhood development" OR "early childhood education" OR "preschool" OR "kindergarten" OR "early education" OR "young children" OR "toddler" OR "infant" |
| 4 | 1 AND 2 AND 3 |
| 5 | Checking through the reference lists of the relevant published reviews |

Note: The asterisk "*" was used as a wildcard symbol appended at the end of the terms to search for variations of those terms.

A total of 2664 articles was screened after the Boolean operation in the selected databases, and the full texts of 256 articles were downloaded after reviewing the titles/abstracts, removing the duplicates, and adding the papers by checking through the reference lists of those articles that did not meet the inclusion criteria. Finally, 22 articles were identified for inclusion after reviewing the full text and excluding the irrelevant studies with the specified reasons listed in Figure 1.

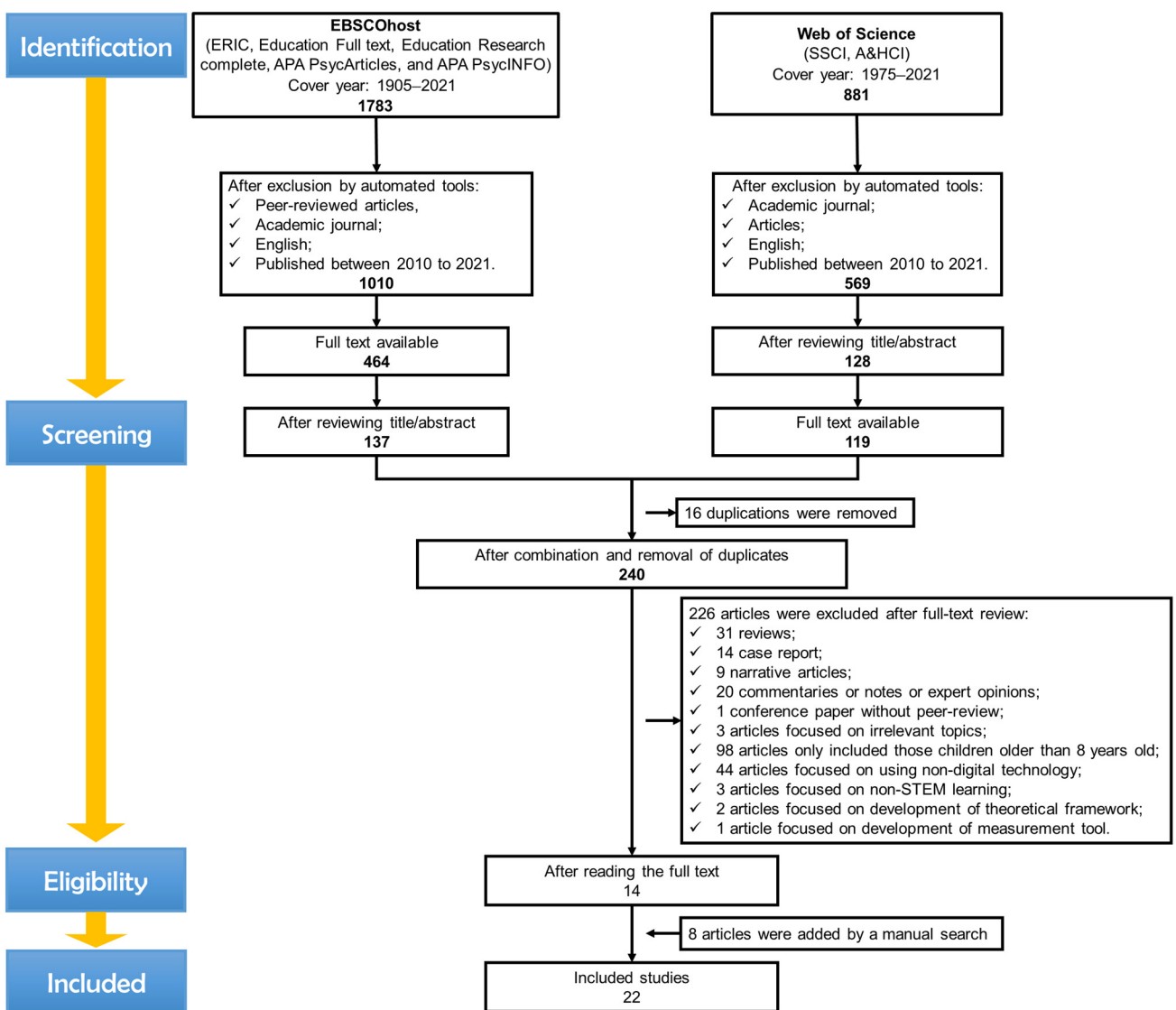

**Figure 1.** Preferred Reporting Items for Systematic Reviews and Meta-Analyses (PRISMA) flow diagram. Search process for selecting studies about the effect/roles of digital technology-aided STEM education in children at a young age.

## 2.2. Selection Criteria

### 2.2.1. Inclusion and Exclusion Criteria

The included articles were (1) original studies; (2) published in peer-reviewed academic journals; (3) written in English; (4) using quantitative, qualitative, or mixed measures; and (5) reported effects or outcomes of the pedagogical intervention programs that used digital technology for assisting integrated STEM education in children in the early years. An article was excluded if it focused on (1) children older than eight years; (2) an approach without digital technology; (3) content unrelated to young children's STEM education; (4) learning a single discipline of STEM; or (5) it is reported by review articles, commen-

taries/notes/editorials, or expert opinions; (6) it is for the development of a theoretical framework; or other irrelevant topics.

### 2.2.2. Screening Process

Two authors conducted article screening. During the process, a consensus by comparison or discussion was made between the authors. After exclusion by the automated tool of databases, titles/abstracts of 1579 articles were screened. The titles and abstracts of the articles were screened to determine their relevance to the focus of this review. After title/abstract screening, the remaining 240 articles were read in whole to determine if they met the inclusion criteria. Another eight research papers were included by checking through the reference lists of the relevant published literature. Finally, 22 relevant articles were identified.

### 2.2.3. Quality Assessment

The study quality of the included articles was assessed according to a reported structured questionnaire [48]. This tool evaluates multiple facets of a study from nine aspects. The total score of the scale ranges from 9 to 36, while the mean score is 22.5. The criteria (good, fair, poor, and very poor) with detailed ratings descriptions are shown in the Supplemental Material. Two authors independently completed the quality assessment of the study. The structured frame, i.e., study design, samples, and key measurement(s), were used to assess the eligible studies, as shown in Supplemental Material Table S1.

### 2.2.4. Data Extraction and Analysis

Two authors independently extracted the data. Any discrepancies in the results were resolved through consultation with a third independent reviewer. A consensus was reached via discussions. The authors of this review first assigned keywords to identify the design and outcomes of the pedagogical programs using digital technology to assist integrated STEM education in children in the early years. Secondly, any relevant outcomes were extracted from both quantitative analyses (e.g., intragroup or intergroup differences, etc.) and qualitative interviews (e.g., extracted themes). Thirdly, data extraction from the included studies was achieved by a thematic approach via a standardized table (see Table 2). This table provides the study design, place where the was study conducted, basic information of participants, technology involved in STEM education, the duration of the STEM program, and role(s) technology played in young children's learning and sensemaking (according to the specific definition/criteria of each category or step published in the literature).

**Table 2.** Key information of the included studies.

| No | Included Studies /Country /Study Design | Samples | Involved Technology & the Duration of the STEM Program [a] | Role Technology Played in the Young Children's STEM Education [b] | Improvement in Young Children's STEM Education | |
|----|------|---------|------|------|------|------|
| | | | | | STEM Knowledge or Skills Acquisition | Engagement in STEM Education |
| | | | | Children with normal development | | |
| 1 | US Randomized controlled trial [49] | 96 Grade I children (aged 6 years, 48 boys and 48 girls) | R, P, M: 20 min (single bout, "pet" robot, screen required) | W | NA | Interest, motivation, enjoyment, self-efficacy |
| 2 | US Pre-/post-test [50] | 34 Children (aged 4.5–6.6 years, 23 boys and 11 girls): 10 pre-kindergarteners, 24 kindergarteners | R, P, M: 20 h (CHERP programming software and LEGO robotics kits, screen required) | W, T | Sequencing | NA |
| 3 | US Quasi-experiment [51] | 42 Children (pre-kindergarteners to kindergarteners, 20 boys and 22 girls): 29 children (intervention); 13 children (comparison) | R, P, M: 1 week, total 10 h (CHERP programming software and LEGO robotics kits, screen required) | W, T | Sequencing | NA |
| 4 | US Pre-/post-test [52] | 53 Kindergarteners (28 boys and 25 girls) | R, P, M: at least 20 h for 6 classes (TangibleK curriculum, screen required) | W, T | Debugging, concepts of robotics, no gender gap | NA |
| 5 | Australia Post-test [53] | 16 Primary school students (aged 5.5–7 years) | R, P, M: 6 weeks, 1–1.5 h/week (laptop and LEGO kits, screen required) | W, T | Numeracy skills, literacy skills | Motivation, communication, collaboration |
| 6 | Australia Quasi-experiment [54] | 135 Year II primary school students (70 boys and 65 girls): 40 students (intervention); 95 students (comparison) | R, P: 6 weeks, 45 min/week (*Scratch* coding software and coding robots, screen required) | W | Patterning, coding | NA |
| 7 | Italy Post-test [55] | 389 Students: 178 Year I-II primary school; 62 Year IV-V primary school; 149 lower secondary school | R, P: 8 weeks, 2 h/week (Bee-Bot and LEGO robotics kits, no screen required) | W | Robotic knowledge, no gender gap | NA |
| 8 | US Quasi-experiment [4] | 105 Children (kindergarteners to Year V primary school students): 48 children (intervention); 58 children (comparison) | R, P: 7 weeks, 1 h/week (KIBO robotic kits, no screen required) | W, T | Numeracy skills, no gender gap | Interest and intention (no gender gap) |

**Table 2.** *Cont.*

| No | Included Studies /Country /Study Design | Samples | Involved Technology & the Duration of the STEM Program [a] | Role Technology Played in the Young Children's STEM Education [b] | Improvement in Young Children's STEM Education | |
|---|---|---|---|---|---|---|
| | | | | | STEM Knowledge or Skills Acquisition | Engagement in STEM Education |
| 9 | Singapore Post-test [56] | 98 Children (aged 3–6 years) | R, P: 7 weeks, 1 h/week (KIBO robotics kits, no screen required) | F, W, T | Programming concepts, sequencing, creativity | Content creation, communication, collaboration |
| 10 | [c] Italy Stepped Wedge randomized trial [57] | 12 Children (aged 5–7 years, 5 boys and 7 girls) | R, P: 6 weeks, 13 classes, 75 min/class (the Bee-Bot robot, no screen required) | W | Coding skills | NA |
| 11 | US Post-test [58] | 60 Children: 15 pre-kindergarteners, 18 kindergarteners, 16 first graders, 11 second graders | R, P: 8 weeks, 1 h/week (CHERP programming software and KIWI robot, screen required) | F, W, T | Robotic knowledge, coding skills, and age differences were noted | NA |
| 12 | US Pre-/post-test [59] | 37 Pre-kindergarteners (20 boys and 17 girls). | R, P: 5 days, 2 h/day (CHERP programming software and LEGO robotics kits, screen required) | W, T | Knowledge of engineering and robotics, building robots | Engagement in math and literacy |
| 13 | US Post-test [60] | 7–20 Elementary schools, 6–29 middle schools | R, P: 6–8 weeks, 1–5 times/week, 30–90 min/time (LEGO robotics kit, no available information on if a screen was required) | W, T | Problem-solving skills | Attitude, collaboration, self-esteem, motivation |
| 14 | US Post-test [61] | 31 Parents and their children (aged 4.5–5 years, 15 boys and 16 girls) | P, M: 20 min (single bout, PBS KIDS *ScratchJr* apps, screen required) | W | NA | Question-asking talks in parent-child interaction |
| 15 | US Pre-/post-test [62] | 28 Children (aged 4–6 years, 14 boys and 14 girls) | P, M: 5 days, 3 h/day (*Daisy the Dinosaur* and *Kodable* apps, screen required) | W | Concepts of programming, sequencing, no gender gap | Enjoyment |
| 16 | Canada Randomized controlled trial [63] | 13 Children (aged 4–5 years, 7 boys and 6 girls): 7 children (intervention), 6 children (control) | M: 10 days, 20 min/day (mathematical apps, screen required) | F | Numeracy skills, difference in ability were noted | Attention, interest |
| 17 | US Factorial design randomized trial [64] | 44 Preschoolers (aged 3–5.5 years, 27 boys and 17 girls) | M: 30 min (single bout, quantity game *Don's Collections* and growth game *Life Cycles*, screen required) | F | Knowledge transfer skills, differences in age and learning condition (i.e., playing or watching) were noted | NA |

Table 2. *Cont.*

| No | Included Studies /Country /Study Design | Samples | Involved Technology & the Duration of the STEM Program [a] | Role Technology Played in the Young Children's STEM Education [b] | Improvement in Young Children's STEM Education | |
|---|---|---|---|---|---|---|
| | | | | | STEM Knowledge or Skills Acquisition | Engagement in STEM Education |
| 18 | US Randomized controlled trial [65] | 60 Preschool children (aged 3–6 years, 25 boys and 35 girls): 20 children (interactive game touchscreen tablet), 20 children (non-interactive video), and 20 children (non-STEM game control) | M: 3 trials of the game (stimulus game *Measure That Animal*, screen required) | W | Knowledge transfer skills, differences in age and learning condition (i.e., playing or watching) were noted | NA |
| 19 | US Randomized controlled trial [66] | 62 Pre-K children (aged 4.8 ± 0.42 years, 26 boys and 36 girls): 2 classes (intervention), another 2 classes (control) | M: 10 weeks, 30–45 min/week (*Creative Curriculum* for science, educational game apps for technology, engineering activities for engineering, planned math activities and lessons for mathematics, screen required) | F | Mathematic knowledge, numeracy skills, tasks of science and engineering | Engagement, communication, collaboration |
| 20 | US Pre-/post-test [67] | 115 Children (aged 5–8 years, 57 boys and 58 girls) | M: 4 weeks (*FETCH!* television episodes, online multi-source, and offline hands-on activities, screen required) | W | Basic physical scientific knowledge | Use pattern, visiting frequency, enjoyment, attitude |
| | | | **Children with special education needs** | | | |
| 21 | US Single-subject research design [68] | 3 Caucasian children diagnosed with Down syndrome: 4–7 years, 2 girls and 1 boy | R, P, M: 5 sessions (*Dash* robot, physical coding blocks, coding apps *Blockly*, screen required) | W | Coding skill | Enjoyment |
| 22 | Australia Single-subject research design [69] | 1 Boy aged 5.5 years and diagnosed with autism | M: 7 sessions (video clips, screen required) | F | Numeracy skills | Self-esteem, communication |

[a]. Involved technology: R = robotics, P = programming, and M = multimedia. [b]. How the technologies served as tools for learning: F = learning *from* technology (defined as *"when technology serves as a storage and delivery tool of knowledge for learners, which limits learning to the content carried by the technology"*, Jonassen et al., 1998 [43]); W = learning *with* technology (defined as *"when learners use technology as a cognitive tool to access, organize, interpret and analyze information, which allows learners to obtain the new knowledge actively and is unlimited to the knowledge stored in the technology"*, Jonassen, 1995 [44]; Jonassen et al., 1998 [43]), and T = learning *through* technology (defined as *"when instructors or learners use technology as a platform to generate new and/or useful technology from the original technology, which may create a technology-enhanced learning environment for teachers and learners"*, Yuan et al., 2019 [45]). [c]. The study focused on executive function and mental health after receiving digital technology-aided STEM education.

### 3. Results

*3.1. Overview of the Included Studies*

The PRISMA flow diagram is presented in Figure 1. Twenty-two papers published between 2010 and 2021 were included in the final analysis, while the complete list and the key information of the included studies are shown in Table 2. These studies reported the effects or roles of digital technology-aided STEM education intervention programs in children who were in the age range from pre-kindergarten to Year III primary school. The participants were from the US (14), Australia (3), Italy (2), Singapore (1), UK (1), and Canada (1). Among the studies, six were post-tests, five were pre-/post-tests, three were quasi-experiments, six were randomized control trials (i.e., parallel design, factorial design, and Stepped Wedge design), and two were single-subject research design (SSRD) studies. Furthermore, the studies were measured by quantitative or qualitative measures, including video observations, in-depth interviews, questionnaire surveys, time to complete task or percentage of task completion in a specific time, academic examinations, etc. Commonly, the sample size was relatively small in the included studies. Only five studies had more than 100 children participants, one study reported the number of recruited schools instead of participating students, and two SSRD studies reported individual participants at numbers of one and three. In regard to the duration of programs, three studies used a single about of STEM education (20–30 min) [49,61,64], and other programs were implemented as chronic practice with a 1–10-week curriculum.

*3.2. Quality of Included Studies*

The total score of study quality ranged between 18–35 for the included studies, whereas 82.4% of studies (14/17) had a total score higher than the mean score of 22.5 based on the questionnaire of Hawker et al. [48]. This indicates the relatively high quality of data provided by the reviewed studies. Studies with obvious sampling shortcomings in sampling, ethics and bias, and transferability received lower scores based on the criteria. This was frequently related to a small sample size and/or a non-experimental design.

*3.3. Types of Digital Technologies Used in the Included STEM Programs*

- Robotics

In the included studies, 14 programs used robotics to assist young children's STEM education. The robotics curriculums all incorporated program design, among which six included multimedia. Master et al. [49] report that participating children used smartphones to control the robots they had programmed, whereas computer devices were used to program the robotics kits in the intervention group of another three studies [50–52]. In the STEM integrated curriculum, the approach was applied and incorporated with different technologies, such as robotics, architecture, laboratory, National Air and Space Museum, etc. Taylor [68] reports an attempt to help children with intellectual disabilities learn computer programming skills and problem-solving strategies. After the intervention phase, their independent coding skills using an iPad application were tested in a one-on-one setting.

Regarding the content of the robotics programs, Sullivan et al. use the concept of relevant tools. This includes common tools, engineering design processes, engineers, robots, and programming. They then look at how participating children complete a variety of tasks. These tasks mainly targeted knowledge and the practice of engineering design and mathematical thinking (including patterning, sequencing, measuring, special thinking, logic thinking, etc.) [4,50–52,56,58,59]. The frequently used robotic kits or systems in the included studies were LEGO robotics kits (e.g., Mindstorms NXT kits, WeDo kits), KIWI robotics kits, KIBO robotics kits, and Bee-Bot/Pro-Bee-Bot.

- Programming

Sixteen studies involved program design as the digital technology to help STEM education in young children. Programming was incorporated with either robotics or other media. The widely used programming software or kits in the included studies were KIBO's

tangible programming blocks, the CHERP program, LEGO Mindstorms program, and Scratch game series. Programming was also integrated as the function of some applications on touchscreen devices for STEM education. In a recent study, participating children used and connected the iPad game "PBS KID ScratchJr" coding buttons to let game characters perform a sequence of actions [61]. In another current study reported by Pila et al. [62], twenty-eight kindergarteners learned coding via the iPad apps named "Daisy the Dinosaur" and "Kodable".

- Multimedia

Fourteen studies involve multimedia, six of which use multimedia alone to help young children learn STEM. Aladé et al. [65] and Schroeder and Kirkorian [64] tested the effect of interactivity with children in iPad game-aided learning of mathematic knowledge (i.e., iPad games named "Measure That Animal", "Don's Collections", and "Life Cycles"). In these games, children were asked (1) to use objects to measure the height or length of the animals; (2) to collect, organize, present, and compare the data in a bar chart; or (3) to put organisms in the sequence of youngest to oldest. Moreover, Miller [63] used 15 iPad apps to increase the numeracy skills of young children at 4–5 years. Similar to these two studies, Aldemir and Kermani [66] also used iPads as a part of a comprehensive intervention package to deliver knowledge or skills in science, engineering, and mathematics to young children. Paulsen and Andrews [67] used transmedia technology using TV and video together with an online game, "Spyhounds", to enhance scientific-related knowledge in young children. Jowett et al. [69] used an iPad-based video to increase the numeracy skills of a 5.5-year-old with autism.

### 3.4. Roles Digital Technology Played in the Included STEM Programs

Following the criteria used in young children's STEM education, the roles of digital technologies were analyzed in each of the included studies, encompassing both single and composite roles. As outlined in Table 2, certain studies demonstrated that digital technology played a singular role in children's STEM education activities, such as learning from or with technology. Six studies [63–67,69] showed that participating children acquired STEM knowledge from or with multimedia (e.g., apps/devices/video clips to store and carry the knowledge). Seven studies [49,54,55,57,61,62,68] indicated that young children learned STEM knowledge/skills with a package of technologies (i.e., robotics + programming + multimedia, robotics + programming, or programming + multimedia) or technology-integrated multi-activities.

Several other studies have shown that digital technologies play composite roles in children's STEM education. Seven studies [4,50–53,56,59,60] indicate that young children acquire STEM knowledge or skills through a package of technologies. Additionally, two studies [58,59] show that the roles of digital technology changed dynamically based on the stages of young children's STEM education. For instance, children initially learned the culture of their community or nursery rhymes from multimedia. They then acquired STEM knowledge or skills through robotics, programming and multimedia. Finally, they learned through the package to construct new robots capable of navigating their community map or dancing to the nursery rhymes they had learned.

### 3.5. Outcomes of the Included STEM Programs

The results of the included studies focused on the short-term effectiveness of digital technology-aided STEM education programs. Effectiveness was demonstrated by young children's improvement in STEM knowledge or skill acquisition and their engagement in STEM education. To conduct the analysis, we (1) assessed the enhancement of STEM knowledge or skill acquisition in STEM-related the disciplines and skills, and (2) interpreted the improvement in STEM education engagement based on indicators of learning engagement, encompassing cognitive, behavioral, and emotional aspects. This meta-construct is widely used in the literature to understand the educational psychological change in

learners who experienced digital technology-mediated learning [70–72]. The key findings from the included studies are summarized in Table 2.

- Improvement in young children's acquisition of STEM knowledge or skills

Twenty studies assessed participating children's STEM knowledge or skills after the programs. Among these studies, eleven reported improvements in science-related knowledge or skills, eight reported improvements in technology-related knowledge or skills, ten reported improvements in engineering-related knowledge or skills, and fifteen reported improvements in mathematics-related knowledge or skills.

In regard to the improvement of science, children were assessed to (1) have an improvement in awareness of scientific knowledge [62,65,66], (2) be familiar with the structures and functions of common tools [59] or objects [54], basic robotic parts [55,58,68], and fundamental concepts of programming [4,52,58], and (3) receive a clear understanding of the terminology and mechanisms behind the tasks [53].

Combining the adoption of robotics and programming has been shown to enhance children's concepts, knowledge, and skills in technology. For instance, children have the opportunity to learn about new technologies, such as robotics [54,55], and develop an understanding of fundamental programming concepts required to operate robots [56,62]. Moreover, during interactive game activities involving robots, children engage in hands-on programming by creating or modifying simple code [57,58,68]. Additionally, Castro et al. [55] reported a significant improvement in the scores of all participants on the Robotics Questionnaire following educational robotics activities conducted by the researchers ($p$ = 0.000). Furthermore, researchers have also observed that integrating programming and multimedia can enhance children's understanding of technology, such as tablet applications with a coding game like Daisy the Dinosaur and Kodable [62].

The combination of robotics and programming has also been found to enhance children's knowledge, concepts, and skills in engineering within STEM activities. The acquisition of basic engineering concepts, understanding of the engineering design process, and the ability to construct and/or program robots to solve problems or complete challenging tasks are important indicators of engineering learning quality. In addition to increasing basic knowledge of engineering and engineers [4,52], the studies provided evidence to show that young children can (1) design, build, and program the robotics [4,52,55,57–59], and (2) transfer the observed geometric information into computer coding [54]. Taylor [68] also reports an evidence-based intervention approach to achieve STEM education in young children with intellectual disabilities. They used explicit instruction, concrete manipulatives, and tangible interfaces, all supported by digital technology. Although such children had difficulty generalizing skills to tablet applications, they still showed a preliminary chance to program the robot.

Moreover, the development of mathematical skills plays an important role in STEM education. After digital technology-aided STEM education, young children made significant progress in overall mathematics ($p$ = 0.003) [66], numeracy skills [4,53,63,66], patterning ($p$ = 0.001) [63], sequencing ($p$ < 0.05) [50,51,62], and measuring [65]. Learning of spatial thinking [57] and logic thinking (e.g., condition statement) were also enhanced [4,52]. For young children with special education needs, an iPad-based VM package successfully taught basic numeracy skills to a 5-year-old boy with autism (i.e., he was able to identify, write, and comprehend the numbers one–seven), and he acquired the stable maintenance of this skill [69].

- Improvement of young children's engagement in STEM education

Thirteen studies reported that learning engagement improved in young children. Three dimensions were identified, including cognitive, behavioral, and emotional engagement.

Firstly, cognitive engagement was increased during the digitally supported STEM education. This was evident in improved understanding [4,52,53,59,66], increased problem-solving behavior [64,69], developed literate thinking [53,59], and better self-regulated interest [49,53,60].

Secondly, improved behavioral engagement was shown in STEM education, as indicated by elevated numbers of participation [60,67], more involvement with learning objects [49,59,66], enhanced sustained and selective visual attention [57,63], improvement in behavioral control [57], and increased on-task behavior and task engagement [59,61,65].

Finally, participating children were more emotionally engaged. This was observed from multiple aspects, including increased enjoyment, fun, and interest when the children were using digital technology in STEM [49,62,67], and improvements in collaborative social interaction and interpersonal skills (i.e., parent-child interaction [61], peer interaction [4,53,57,60,66], and teacher-student interaction [59]).

- Factors that potentially affected young children's technology-integrated STEM education

Three studies investigated the gender gap in young children's STEM education, and no such gap was found in any of them [49,52,62]. Master et al. [49] investigated whether positive experiences with programming robots would lead to greater interest and self-efficacy among girls, despite gender stereotypes in STEM education. The result showed that 1st grade female elementary children with programming experience reported higher technology interest and self-efficacy than those without this experience. The data did not exhibit a significant gender gap relative to participating boys' interests and self-efficacy. Pila et al. [62] indicated that no relationship was found between the gender of preschoolers and their outcomes in learning code via tablet applications. In the study by Sullivan & Bers [52], no significant gender difference was found in accomplishing tasks of the *TangibleK Robotics Program*. This implies that kindergarten boys and girls were equally successful in building and programming.

Age-appropriate integration is also related to young children's STEM education. Castro et al. [55] indicate significant improvements in engineering and technological concepts for all ages of participants who attended the educational robotics introduction program (even lower grades of primary schools). This may be a potential predictor of performance on the knowledge transfer tasks tested in the other two studies. Such tasks assess children's adaptive skills that transfer what they have learned in one context to another [73]. Schroeber & Kirkorian [64] indicate that the effectiveness of STEM education via touchscreen games varies by age and learning conditions (i.e., playing or watching). Younger children (aged 3.04–4.29 years) learned from the quantity game only by watching the game (i.e., they obtained higher scores in the direct post-test assessment than the pre-test, $t(12) = 3.21$, $p = 0.008$, $d = 0.90$). However, information learned from the quantity game was not transferred. However, older children (4.39–5.41 years) learned from the growth game in both conditions (i.e., $t(12) = 3.67$, $p = 0.003$, $d = 1.07$ in the watching condition and $t(8) = 3.24$, $p = 0.012$, $d = 1.15$ in playing condition). In addition, older children's learning is generalized to near transfer tasks in both conditions, but their learning is only generalized to far transfer tasks in the watching condition.

Playfulness and interactivity are other crucial factors influencing young children's technology-integrated STEM education. Aladé et al. [65] found that participants demonstrated more significant knowledge transfer than the control condition, whether they were playing with an interactive tablet-based game or watching a non-interactive video. In particular, participants in the playing condition performed better on near transfer tasks. Several additional studies [51,62–64,67] have also shown that incorporating fun, playful, and interactive content or functions in digital technology enhances children's interest, motivation, and positive acquisition of STEM-related concepts, knowledge, and skills.

### 3.6. Participants' Acceptance and Satisfaction with the Included Programs

An overall satisfaction/acceptance was observed in participating children and teachers of eight included studies, which reported the outcomes of process evaluation. Five studies asked the children about their attitudes towards the digital technology-aided STEM education programs. Children thought the activities were enjoyable and expressed that they were happy, fun, and interested in the program [49,59,60,62,67]. Participating children

in two studies stated an increased self-efficacy and self-confidence in STEM education after they accomplished the tasks of the programs [49,60].

Five studies reported that teachers thought digital technology aided children's STEM education [4,53,59,60,66]. According to the interview written by Aldemir and Kermani [66], teachers thought that the digital technology programs helped them integrate and implement STEM-related activities in a developmentally appropriate manner. It enabled them to scaffold and help children practice previously learned knowledge in hands-on, three-dimensional activities. Sullivan and Bers [4,56] report that teachers had the independence and confidence to adjust to digital devices. In other studies, teachers emphasized the efficacy of digital technology in children's improvements in literacy and numeracy skills [53,59], interpersonal skills [53,56,60,66,69], and skills of critical thinking [60].

### 3.7. Acknowledged Limitations in the Included Studies and Implications for Research Design

The limitations in the included studies are highlighted by the assessment of study quality (Table S1). Based on the summary of the included studies, half of the studies were conducted with an experimental design (including randomized trials, quasi-experiments, and single-subject research design), whereas another half used a non-experiment study design. As such, the internal validity is relatively low for half of the evidence. Pedagogic intervention programs predominantly use cluster randomization sampling to decrease possible contamination between the subjects. Again, the inactive control group used in some studies may only weakly support the efficacy of STEM learning programs since participants in the control group received no activity [49,51]. As such, it is hard to say whether the different achievements between groups can be attributed to digital technology-aided STEM education.

Half of the included studies reported a small sample size, even though some claimed to be randomized controlled trials. This may hinder the external validity of such intervention programs. Policymakers or stakeholders should be made aware of these developments so that they may provide more resources for future trials. A shortage of resources for intervention and a limited time of observation may be related to the attrition of participants and the limited resources that the research team could access. These limit the depth and breadth of the analysis. Furthermore, the nonrandomized selection of the sampled children (relevant to the school arrangement or volunteering of the students), differences in teachers' interest, and the education level of parents may have influenced the adherence to the study program/intervention, which resulted in the possibility of selection bias and/or subjective bias in outcomes. The nature of young children also requires more time for intervention and assessment design.

In addition, methods of evaluating its effectiveness are limited as follows: (1) measuring instruments may not be standardized or validated; (2) the assessments or instruction may be completed by teachers who may have bias or mistakes during the process; and (3) the assessments were selected as intent to treat, which may not useful to generalize knowledge/skills from research to other settings (e.g., transfer task to test problem-solving skills).

Given the consolidated framework of implementation science [74], it is important to include the attitudes and adoption of parents and school principals in the process evaluation because they are important members of the personal community and course community for supporting students in learning engagement. A review states that the interviewed parents presented a highly positive attitude toward supporting STEM education in young children [42]. However, kindergarten and school principals still lack information in the literature to advise them on this topic. Finally, the situation in a real-world practice may be different from the research, where a cultural/societal difference or other confounders may also interfere with the effect.

## 4. Discussion

This scoping review analyzed and summarized the data of empirical studies to investigate the role and effect of digital technology on supporting STEM education among young children. Results showed the different roles and degrees that digital technology played in young children's STEM education. A relatively positive outcome was presented by improvements in STEM knowledge/skill acquisition and engagement in STEM education. The featured factors were additionally considered and analyzed. Moreover, in the following session, the difficulties, challenges, and criticisms of using digital technology were highlighted, and the implications for educational practice and research design were discussed.

### 4.1. Theoretical Mechanisms Embedded in the Evidence

According to the role digital technology played in the learning process, robotics and programming successfully helped young children to (1) learn the basic STEM concepts from technology, (2) learn STEM knowledge/thinking/skills with technology, and (3) create new functional robots or program through technology.

However, it is evident that multimedia used in STEM education focused more on transferring existing knowledge to children than providing children with the opportunity to organize, interpret, and analyze the information; as a result, the single use of multimedia in the included studies mainly played the role of learning from technology.

This review revealed that digital technology was capable of motivating young children's engagement in STEM education, which was in line with the theoretical framework "Academic Communities of Engagement" [70]. The included studies showed that digital technologies presented STEM knowledge and information in a way that learners enjoyed and found interesting. Positive experiences in STEM classes implied a strong engagement in STEM education among children, though this may be associated with success in assigned tasks [75]. During STEM education, self-motivation and self-confidence increased in children, and they wanted to be challenged with new tasks. Subsequently, tasks became more difficult as the children completed them. This process drove children to seek help from their teachers, peers, or parents, which in turn enhanced their social interaction skills and problem-solving behaviors. It was found that young children's literacy and numeracy skills significantly developed, which indicates an increased understanding of knowledge and thinking. The cognitive progress promoted self-efficacy and self-esteem in the children, which increased their positive emotions in turn.

Multiple technologies-assisted STEM education had a significant effect size on some indicators of cognitive development in early schoolers based on the Cognition–Priming Model [76]. This is also supported by the evidence included in this review. The executive function of the pilot 5- to 6-year-old children (i.e., both visuo-spatial working memory and inhibition skills) was significantly improved after participating in a 6-week program incorporated with educational robotics [57]. This implies that digital technology may potentially have a positive effect on the cognitive development of young children.

### 4.2. Featured Factors and Implications

Based on the current review, no gender gaps were found in the effectiveness of digital technology-aided STEM education in young children [4,49,52,55,62]. This indicates that digital technology may help both boys and girls have positive experiences in STEM education and increase their self-efficacy. This may overcome public stereotypes of gender and potentially increase the possibility for girls to choose careers in science.

This review provides evidence that digital technology is effective and beneficial to children at a young age for their STEM education. Technologies used in the reported programs were all effective in teaching STEM knowledge to young children. Programs were efficient and took effect in just a few weeks. The studies analyzed in this review showed that young children could create new technology through original technology, even though most were preschoolers [4,58,59]. However, the ability to transfer knowledge was influenced by

age and learning conditions (playing or watching a digital device) [64,65]. This observation implies that (1) younger children may learn from technology but temporarily have no adaptive skill of transferring knowledge to another context, while (2) older children may also learn from technology and perform better in far transfer tasks with the watching condition. In turn, they may accomplish near transfer tasks if they play with the technology. Thus, a well-planned curriculum with age-appropriate technology usage is crucial for different learning targets in young children's STEM education.

Digital technologies also play a significant role in integrating STEM subjects across disciplines. Of the 22 papers reviewed, 12 demonstrate that activities based on digital technology take diverse forms, covering concepts, knowledge, and skills across multiple disciplines. These activities encompass two disciplines [57], three disciplines [4,52,53,55,56,62,66,68], and four disciplines [54,58,59]. Most projects (n = 11) employed a combination of digital technologies, such as robotics, programming, and multimedia, or robotics and programming. Teachers selected the technology package based on children's abilities and learning progression. Different digital technologies were introduced at various stages of learning to support children's development of STEM-related concepts, knowledge, and skills. As children's capabilities increased, the selection of technology types or pedagogy evolved, and the complexity of related operations and learning tasks also increased—transitioning from learning digital technology through operation demonstrations to being able to critically evaluate and predict changes in the motion trajectory of digital technology, such as robots. This evidence suggests that technology-integrated STEM education can facilitate integrating teaching and learning processes into a holistic and continuous approach rather than fragmented and intermittent learning [36,37].

Among the digital technologies adopted in the included studies, robotics was the most popular and effective (n = 10). Through interactions with robots, children progress from understanding the robot's structure [52–55,58,59] to enhancing the robot's abilities using programming skills [54–58,60]. Eventually, they use robots to bring their ideas to life [4,53–55,57,59]. These activities involve various STEM disciplines and even extend to other disciplines, such as literacy [53] and art [56]. This effectively addresses the lack of discipline-specific content in young children's STEM education [31,34], such as engineering [32,33], and helps to balance the disciplinary emphasis [31–35].

Since digital technology is taken as an appropriate learning mode for young children, regardless of culture and ethnicity [53], it provides an opportunity to increase the interest and engagement with STEM knowledge/skills among those young children who are from underrepresented groups (e.g., ethnic minority or low socioeconomic status) mboxciteB60-education-2759536,B66-education-2759536; in addition, it showed equal effectiveness in the participating children with diverse backgrounds [53]. Digital technology helps young, ethnically diverse children to learn about each other's cultures in their STEM class [4]. As such, digital technology-aided STEM education improved the balance of educational equity in young children with diverse cultures, ethnicities, and socioeconomic statuses.

### 4.3. Difficulties/Challenges and Criticisms & Implications for Educational Practice

There are considerable difficulties and challenges in using digital technology to support young children's STEM education. First, as in a previous systematic review [77], digital technology may only be a carrier of knowledge, promoting rote learning of numbers/letters and a lack of deeper conceptual understanding and thinking skills. Second, the difference in age learning condition existed [51,64], which suggests a rationale for digital technology in planning STEM curriculum. Third, it was demonstrated that pre-kindergarteners were only able to master some basic robotics and programming skills. At the same time, older children were able to master advanced concepts and tasks under the same circumstances [4,58]. Significant scaffolding, structured guidance, and attention should be paid to very young children when they learn to master new concepts [59]. This implies that one-on-one adult help is necessary, whereas the recommended ratio of students to instructors is 4:1.

Moreover, high-quality interactions between children are also required. It was found that task-relevant talks improved the children's learning of coding, while a large proportion of question-asking talks decreased the efficiency of the children's learning of coding [64]. Nevertheless, the authors argued that the large proportion of question-asking talks indicates participants' difficulty. Question-asking may encourage problem-solving skills and extend the learning process over time, and this was not shown in the short-term effect. In addition, Miller [63] found that young children may abandon or use trial-and-error attempts to find the correct answers when they use an app with difficult challenges, in which they use memory rather than skills to make progress. Finally, apps featuring brightly colored decorations or animated characters may capture children's attention. However, they may not intensely engage with apps aimed at skill development [63]. Digital technology and interactive media warrant appropriate development and reasonable utilization in early childhood settings [78].

### 4.4. Limitations of this Review

Aside from the fact that some of the included studies possessed limitations that may have affected the findings of the present review, this work also has limitations. To emphasize and focus on the multi-/inter-disciplinary nature of STEM learning, this scoping review initiated this topic with a relatively narrow search term (i.e., "STEM"). This may only cover the tip of the iceberg by including articles explicitly mentioning "STEM" learning. Similarly, articles were retrieved from seven electronic databases, and only those including the specific terms mentioned in the title or abstract were reviewed for further analysis. As a result, future reviews may extend the search terms from including "STEM" only to all single disciplines and investigate and discuss in a broader scope. Furthermore, studies that were not in English; published in conference abstracts, government reports, or textbooks; or unpublished dissertations were not included. The dataset includes the years 2020 and 2021 due to the scope of the study. The subsequent research agenda will include the COVID-19 and following post-pandemic periods to understand the changes. Additionally, cutting-edge technologies (e.g., AI) may be included in the keywords to understand the new trends of digital technologies in early STEM education.

### 5. Conclusions

The present scoping review demonstrated the potential effect of digital technology on young children's STEM education regarding STEM knowledge, skill acquisition, and engagement in STEM education. According to the included studies, digital technology plays different roles in young children's learning. Both children and teachers showed positive attitudes towards the included programs. Based on this review, no gender gap was observed in the participants, but the effect of digital technology may be affected by age and learning conditions. The extracted data also indicated an improvement in the balance of educational equity. Nonetheless, difficulties, challenges, and criticisms of the current situation of using digital technology with young children's STEM education were acknowledged, implying that there are still opportunities for improvements in appropriate development and reasonable utilization of digital technology in early childhood settings. Furthermore, limitations existing in the included studies also need to be resolved in future research. The time frame for this scoping review was limited to the years 2010 to 2021. However, there have been valuable research outcomes in recent years, such as developing and implementing APPs in the STEM project in Australia [79], which showed evidence of connecting digital and non-digital technology in meaningful learning tasks, as well as a comparative study about the Internet of Toys (IoToys) showing the new research focuses of involving technologies and children's agency [80]. The further research agenda aims to incorporate evidence-based data to understand the post-pandemic period and also notice the new trends of emerging technologies, such as AI, in the STEM learning context [81].

**Supplementary Materials:** The following supporting information can be downloaded at https://www.mdpi.com/article/10.3390/educsci14040357/s1, Table S1. Quality assessment of the included papers.

**Author Contributions:** Conceptualization, X.H.; methodology, Y.F. and Y.L.; software, Y.F. and Y.L.; validation, X.H., Y.F. and Y.L.; formal analysis, Y.F.; investigation, Y.F.; resources, X.H.; data curation, Y.F. and Y.L.; writing—original draft preparation, Y.F.; writing—review and editing, X.H. and Y.F.; visualization, Y.F.; supervision, X.H.; project administration, Y.L. All authors have read and agreed to the published version of the manuscript.

**Funding:** This research received no external funding.

**Institutional Review Board Statement:** Ethical review and approval were waived for this study due to that this review was conducted and analyzed based on published data.

**Informed Consent Statement:** Not applicable.

**Data Availability Statement:** The datasets used and/or analysed during the current study are available from the corresponding author (Dr. HU Xinyun) on reasonable request.

**Conflicts of Interest:** The authors declare no conflict of interest.

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
