# Peer review of "Roles and Effect of Digital Technology on Young Children’s STEM Education: A Scoping Review of Empirical Studies"

_education, doi:10.3390/educsci14040357_

Round 1

Reviewer 1 Report

Comments and Suggestions for Authors

This systematic review of empirical studies on the role of digital technology on young children’s STEM learning is very interesting.

Comments and suggestions

·  Line 115. After mentioning ‘Table 1’, once the paragraph is finished, the ‘Table 1’ should be included, in this case, after line 115.

·      Line 121.- In Figure 1 you show the step followed to carry out the systematic review throw PRISMA. You should include the reference from where you have extracted the flow diagram.

·      Line 130.- “The included articles were (1) original studies; (2) ...”; ‘:’ is missed -> “The included articles were: (1) original studies; (2) ...”

·        Line 134.- the same comment as in line 130, ‘:’ is missed.

·        Line 396.- the same comment as in line 130, ‘:’ is missed.

·        Line 266.- the same comment as in line 130, ‘:’ is missed…

·        Lines 102-105: Related with research question 2 and 3 (“(2) What degree the digital technology effective in assisting young children’s STEM learning in terms of acquiring knowledge/skills and learning engagement; (3) Factors have been identified as related to the effect of digital technology on young children’s STEM learning.”) -> suggestion: they could be rewritten to make both clearer to understand.

·      Line 551-552.- “Furthermore, limitations existing in the included studies are also desired to be resolved in future research”. This sentence could be deleted.

·      Exclusion criteria. - All the ‘exclusion criteria’ that have been used (Figure 1), should be justified in the text or included as a limitation (e.g., the fact that you have only analyzed articles written in English -it could be justified because most of contributions in this field are written in English, in case that it is true-), unless a specific criterium is self-explanatory in terms of justification.

·  What is the reason for lust including the articles published between 2010 to 2021 [Figure 1]? Is there a specific reason? … In addition, in lines 99-100 you write: “Thus, the current systematic review was conducted to summarize the key findings from the empirical studies conducted prior to the Covid-19 pandemic” (see also: line 7, in the ‘Abstract’; line 138, in section ‘Results’); but the pandemic was declared on the firsts months of the year 2020 … and in fact all the articles that are listed in the article were effectively published in the pre-pandemic moment.

Then, if it is restricted to the pre-pandemic period, it should include articles until December 2019, and you need to change it in your article; if the review period finishes in 2021 (as written), there are more articles to be included in the review (the newest paper included in the review in from 2019). 

·      Line 164.- In Table 3 -> at the foot of the Table, it is written “c. The study focused on the executive function and mental health after receiving digital technology-aided STEM learning” -> Which items are associated to letter c? Is there anything missing?

·     In section 4.4 you describe the limitations found in the researched articles. Which ones are the limitations of your own article -you should write them-?

·          Section 5.- According to what is written from lines 99-105, the article “summarize the key findings from the empirical studies conducted prior to the Covid-19 pandemic. The research questions included: (1) What role the digital technology play in assisting young children’s learning of STEM knowledge/skills; (2) What degree the digital technology effective in assisting young children’s STEM learning in terms of acquiring knowledge/skills and learning engagement; (3) Factors have been identified as related to the effect of digital technology on young children’s STEM learning.”

o   Line 542: Related to research question 1. Which roles?... In the same line, it could improve the section stating clearly the answers to the research questions.

References

·        Some DOI numbers are missed, e.g.,

-

Karp, T.; Maloney, P. Exciting young students in grades K-8 about STEM through an afterschool robotics challenge. American 677 Journal of Engineering Education 2013, 4, 39-54.

DOI:10.19030/ajee.v4i1.7857

-

Aladé, F.; Lauricella, A.R.; Beaudoin-Ryan, L.; Wartella, E. Measuring with Murray: Touchscreen technology and preschoolers' STEM learning; 2016, 62 (2), pp. 433-441.

 DOI:10.1016/j.chb.2016.03.080

-

Sullivan, A.; Kazakoff, E.R.; Bers, M.U. The wheels on the bot go round and round: Robotics curriculum in pre-kindergarten. 639 Journal of Information Technology Education. Innovations in Practice 2013, 12, 203.

https://doi.org/10.28945/1887

...

Author Response

Reviewer 1"

Please review the attachment, where responses to the reviewers' comments have been highlighted in colour.

RE: education-2759536

Title: Roles and effect of digital technology in young children’s STEM learning: A systematic review of empirical studies

Reviewer 1:

This systematic review of empirical studies on the role of digital technology on young children’s STEM learning is very interesting.

Comments and suggestions

  •  Line 115. After mentioning ‘Table 1’, once the paragraph is finished, the ‘Table 1’ should be included, in this case, after line 115.

A: Thank you for your suggestion. Table 1 was moved in Line 144 of current manuscript.

  •      Line 121.- In Figure 1 you show the step followed to carry out the systematic review throw PRISMA. You should include the reference from where you have extracted the flow diagram.

A: Thank you for your comment. We agree with the reviewer.

In Figure 1, in the “Identification” step, databases selected for search by EBSCOhost and Web of Science are included; they are:

EBSCOhost: ERIC, Education Full text, Education Research Complete, APA PsychoArticles, APA PsychoINFO;

Web of Science: SSCI and A&HCI.

Please kindly refer to Figure 1.

  •      Line 130.- “The included articles were (1) original studies; (2) ...”; ‘:’ is missed -> “The included articles were: (1) original studies; (2) ...”
  • Line 134.- the same comment as in line 130, ‘:’ is missed.
  • Line 396.- the same comment as in line 130, ‘:’ is missed.
  • Line 266.- the same comment as in line 130, ‘:’ is missed…

A: Thank you for your comment. The colon was added to the lines pointed out by the reviewer.

  •       Lines 102-105: Related to research question 2 and 3 (“(2) What degree the digital technology effective in assisting young children’s STEM learning in terms of acquiring knowledge/skills and learning engagement; (3) Factors have been identified as related to the effect of digital technology on young children’s STEM learning.”) -> suggestion: they could be rewritten to make both clearer to understand.

A: Thank you for your comment. The objectives of the study were revised accordingly. Please refer to Lines 110-121.

  •      Line 551-552.- “Furthermore, limitations existing in the included studies are also desired to be resolved in future research”. This sentence could be deleted.

A: Thank you for your comment. The sentence was removed.

  •      Exclusion criteria. - All the ‘exclusion criteria’ that have been used (Figure 1), should be justified in the text or included as a limitation (e.g., the fact that you have only analyzed articles written in English -it could be justified because most of the contributions in this field are written in English, in case that it is true-), unless a specific criterium is self-explanatory in terms of justification.

A: Thank you for your comment. The detailed exclusion criteria are included in the 2.2.1. Inclusion and exclusion criteria. Furthermore, we indicated “Studies that were not in English, published in conference abstracts, government reports, or textbooks or unpublished dissertations were not included.” as limitations of this review in 4.4 Limitation of this review.

  •  What is the reason for lust, including the articles published between 2010 to 2021 [Figure 1]? Is there a specific reason? … In addition, in lines 99-100, you write: “Thus, the current systematic review was conducted to summarize the key findings from the empirical studies conducted prior to the Covid-19 pandemic” (see also: line 7, in the ‘Abstract’; line 138, in section ‘Results’); but the pandemic was declared on the firsts months of the year 2020 … and in fact all the articles that are listed in the article were effectively published in the pre-pandemic moment.

Then, if it is restricted to the pre-pandemic period, it should include articles until December 2019, and you need to change it in your article; if the review period finishes in 2021 (as written), there are more articles to be included in the review (the newest paper included in the review in from 2019). 

A: Thank you for your comment. Indeed, we restricted the studies included in the review to be conducted/investigated prior to the pandemic, that is, December 2019. However, some studies were conducted prior to the pandemic but published after December 2019. Therefore, we did not restrict the published date, but we restricted the study investigated date/period, which was checked in the main text.

  •      Line 164.- Table 3 -> at the foot of the Table is written “c. The study focused on the executive function and mental health after receiving digital technology-aided STEM learning” -> Which items are associated to letter c? Is there anything missing?

A: Thank you for your comment. The word “c” was marked after the reference “Di Lieto et al., 2017” in Table 2.

  •    In section 4.4, you describe the limitations found in the researched articles. Which ones are the limitations of your own article -you should write them-?

A: Thank you for your comment.

Since this systematic review is established based on the existing evidence from previous literature, the limitations indicated by the original studies may cause bias or limitations in the analysis and conclusion of this review. Thus, this content was summarized and shown in Lines 428-457.

The fourth graph of this section (Line 458-466) was the knowledge/research gap detected by this review, which is warranted for future research efforts.

The limitation of this review was in Lines 591-601, which was relevant to the methodology of the systematic review itself.

  • Section 5.- According to what is written from lines 99-105, the article “summarize the key findings from the empirical studies conducted before the Covid-19 pandemic. The research questions included: (1) What role does digital technology play in assisting young children’s learning of STEM knowledge/skills; (2) What degree the digital technology effective in assisting young children’s STEM learning in terms of acquiring knowledge/skills and learning engagement; (3) Factors have been identified as related to the effect of digital technology on young children’s STEM learning.”

o   Line 542: Related to research question 1. Which roles?... In the same line, it could improve the section stating clearly the answers to the research questions.

A: Thank you for your comment. The objectives of the study were revised accordingly. Please refer to Lines 110-121.

References

  • Some DOI numbers are missed, e.g.,

Karp, T.; Maloney, P. Exciting young students in grades K-8 about STEM through an afterschool robotics challenge. American 677 Journal of Engineering Education 2013, 4, 39-54.

DOI:10.19030/ajee.v4i1.7857

Aladé, F.; Lauricella, A.R.; Beaudoin-Ryan, L.; Wartella, E. Measuring with Murray: Touchscreen technology and preschoolers' STEM learning; 2016, 62 (2), pp. 433-441.

 DOI:10.1016/j.chb.2016.03.080

Sullivan, A.; Kazakoff, E.R.; Bers, M.U. The wheels on the bot go round and round: Robotics curriculum in pre-kindergarten. 639 Journal of Information Technology Education. Innovations in Practice 2013, 12, 203.

https://doi.org/10.28945/1887

A: Thank you for your comment. The available information on DOI was added to the reference list.

Reviewer 2 Report

Comments and Suggestions for Authors

Dear Authors,

Upon reviewing your article "Roles and effect of digital technology in young children’s STEM learning: A systematic review of empirical studies," I would like to commend the robust methodology and relevance of your contribution to the STEM learning field in young children. However, I would like to point out a few areas for your consideration:

1.      Currency: While the review is comprehensive, there is a noticeable absence of literature from the past two years. Particularly important is the emergence of Artificial Intelligence (AI), which has significantly transformed the STEM field in education. This evolution represents a significant paradigm shift, and its inclusion could greatly enrich your study's analysis and conclusions. In the past two years, the emergence and rapid evolution of AI have significantly transformed the STEM field in the educational context. This evolution represents a paradigm shift, influencing not only the tools and methodologies employed in STEM education but also the competencies and skills considered essential for students. The omission of this recent literature in your review could limit the understanding of the current and future impact of digital technology on STEM learning. Therefore, discussing how these recent advances in AI might influence the conclusions and recommendations of your study would be very enriching for your work. This observation does not diminish the value of your contribution but suggests a way to ensure its relevance and timeliness in the dynamic field of STEM education. You may, if you wish, highlight this idea in the limitations of the work. 

2.      Discussion of Results: It would be beneficial to deepen the discussion of the results, organizing it according to the specific objectives of the study. This would provide a more detailed and contextualized analysis, further strengthening your work.

These suggestions are aimed at enhancing your valuable contribution to research. Congratulations on your work, and I hope these recommendations will be useful for future research.

Sincerely,

Author Response

Reviewer 2:

Please review the attachment (the reviewers' comments have been highlighted in colour)

Dear Authors,

Upon reviewing your article "Roles and effect of digital technology in young children’s STEM learning: A systematic review of empirical studies," I would like to commend the robust methodology and relevance of your contribution to the STEM learning field in young children. However, I would like to point out a few areas for your consideration:

  1. Currency: While the review is comprehensive, there is a noticeable absence of literature from the past two years. Particularly important is the emergence of Artificial Intelligence (AI), which has significantly transformed the STEM field in education. This evolution represents a significant paradigm shift, and its inclusion could greatly enrich your study's analysis and conclusions. In the past two years, the emergence and rapid evolution of AI have significantly transformed the STEM field in the educational context. This evolution represents a paradigm shift, influencing not only the tools and methodologies employed in STEM education but also the competencies and skills considered essential for students. The omission of this recent literature in your review could limit the understanding of the current and future impact of digital technology on STEM learning. Therefore, discussing how these recent advances in AI might influence the conclusions and recommendations of your study would be very enriching for your work. This observation does not diminish the value of your contribution but suggests a way to ensure its relevance and timeliness in the dynamic field of STEM education. You may, if you wish, highlight this idea in the limitations of the work. 

A: Thank you for your comment. Please allow us to address your concerns for the following reasons.

Firstly, we strongly agree with the reviewer that using AI in STEM education is a very important topic for review, especially those used in remote learning over the past two years.

However, the focus of the current review is the digital technology used in STEM education prior to the pandemic, which may serve as a fundamental situation of technique used in STEM education. For AI use in STEM education, which should be another systematic review with advanced research focus based on this review. Therefore, to lay the foundation, we choose to keep the current review focus from AI use in the past two years. For the current review, we have to address the absence of AI use in STEM education, which is one of the limitations of this review; please refer to 4.4 Limitations of this review.

Finally, it is worthy to conduct a new systematic review on AI use in STEM education. We are therefore inspired by the reviewer's suggestion that “the emergence and rapid evolution of AI have significantly transformed the STEM field in the educational context. This evolution represents a paradigm shift, influencing the tools and methodologies employed in STEM education and the competencies and skills considered essential for students.” This should be the targeted research question of a brand-new review, which may provide a comparison between the AI use and the technique summarized in this review. Eventually, these two reviews may comprehensively address the reviewer’s comment.

  1. Discussion of Results: It would be beneficial to deepen the discussion of the results, organizing it according to the study's specific objectives. This would provide a more detailed and contextualized analysis, further strengthening your work.

A: Thank you for your comment. The discussion session was refined and revised accordingly; please refer to the main text.

These suggestions are aimed at enhancing your valuable contribution to research. Congratulations on your work, and I hope these recommendations will be useful for future research.

Reviewer 3 Report

Comments and Suggestions for Authors

This paper presents a systematic review of empirical articles published between 2010 and 2021 that deal with young children's STEM learning from, with, and through digital technology. I was very glad to have the opportunity to review this manuscript as the topic is quite relevant and would certainly add to existing literature in this relatively new but important area of scholarship.

Unfortunately, I find there is a major flaw in the design of the systematic review in that it only considers articles that look at STEM learning as a whole. This is problematic because STEM is a composite term that includes science, technology, engineering, and mathematics (and often related areas as well). If I'm understanding correctly from the description of the search process, the authors only included articles that included the word "STEM" in addition to some combination of other search terms related to the digital platform and the age of participants. I cannot understand why the authors did not include articles that covered each of the four components of STEM learning (i.e., math learning, science learning, etc.). Perhaps this decision was made to keep the scope of the review manageable. That would be understandable, but the authors need to do a much better job of providing justification for this very narrow, but at the same time conceptually very broad, scope. Indeed, many of the articles do look at just a single STEM discipline (i.e., early math), but because the authors of those studies used the word STEM in describing the context and justification for the study, it was included in this review. The result feels like a very arbitrary distinction for inclusion/exclusion.

I also find the "quality ratings" provided in this paper to be quite problematic. For starters, it is not clear what the purpose of rating (and in effect) ranking the quality, as perceived by the authors, of these studies is. How does it add to the literature to provide these subjective rankings? The scores seem unrelated to the focal results of this paper. If the authors can provide some clearer justification for including these rankings (ideally with some measure of inter-rater reliability), I would also want to see a much more detailed description of what each rating means and how it is determined. What is the practical difference between a 2 and a 3 when it comes to the abstract and title? The sampling? The ethics? What meaning should a reader take away from these ratings?

There are also some organizational issues with this paper, most notably - the text that is currently presented in the limitations section covers limitations of the studies that are reviewed. If comparing these limitations is a focus of the review, this analysis belongs in the results section. The limitations section should be focused on limitations of the current study (the sytstematic review). 

Table 2 and Table 3 both include all 22 studies, but they numbered differently, which makes it very difficult to look at results across the tables.

Section 2.2 Selection Criteria should really come before section 2.1 Search Strategy. Without knowing the selection criteria and the inclusion/exclusion goals, it is very difficult to make sense of the search strategy.

We need MUCH more detail about the search strategy, especially when large swaths of the original sample are removed. For example, at the stage of title/abstract review, what were the authors looking for that would help them determine that an article should be included/excluded? For the 8 articles that were added by hand, why were they added? What did the authors do differently that they wouldn't have come up from the original search strategy?

Comments on the Quality of English Language

The writing of this manuscript needs significant revision. Nearly every sentence (with the exception of the results section, which is significantly better overall, though still includes errors) includes some grammatical errors, awkward word choice, and/or poor sentence structure. The errors do not render the paper incomprehensible, but rather make the writing quite clunky and difficult to follow.

Author Response

Reviewer 3:

Please review the attachment (the reviewers' comments have been highlighted in colour)

This paper presents a systematic review of empirical articles published between 2010 and 2021 that deal with young children's STEM learning from, with, and through digital technology. I was very glad to have the opportunity to review this manuscript as the topic is quite relevant and would certainly add to existing literature in this relatively new but important area of scholarship.

A: Thank you for your comment.

Unfortunately, I find there is a major flaw in the design of the systematic review in that it only considers articles that look at STEM learning as a whole. This is problematic because STEM is a composite term that includes science, technology, engineering, and mathematics (and often related areas as well). If I'm understanding correctly from the description of the search process, the authors only included articles that included the word "STEM" in addition to some combination of other search terms related to the digital platform and the age of participants. I cannot understand why the authors did not include articles that covered each of the four components of STEM learning (i.e., math learning, science learning, etc.). Perhaps this decision was made to keep the scope of the review manageable. That would be understandable, but the authors need to do a much better job of providing justification for this very narrow, but at the same time conceptually very broad, scope. Indeed, many of the articles do look at just a single STEM discipline (i.e., early math), but because the authors of those studies used the word STEM in describing the context and justification for the study, it was included in this review. The result feels like a very arbitrary distinction for inclusion/exclusion.

A: Thank you for your comment. Indeed, it should include articles that cover each of the four components of STEM learning (i.e., math learning, science learning, etc.) from a broad view. However, in the current review, the research focus is the comprehensive and interdisciplinary learning journey that emphasizes the integration of science, technology, engineering, and mathematics, i.e., STEM education (see lines 49-62). Therefore, in order to emphasize the integration of all four disciplines of STEM and ensure the included studies are those including all four components, we used “STEM” as the search term in the search strategy.

In addition, in the section of “4.4 Limitation of this review”, we also stated that “Articles were retrieved from seven electronic databases, and only those including the specific terms mentioned in the title or abstract were reviewed for further analysis.” (Line 593-595) This is one of the limitations of the current review.

I also find the "quality ratings" provided in this paper to be quite problematic. For starters, it is not clear what the purpose of rating (and in effect) ranking the quality, as perceived by the authors, of these studies is. How does it add to the literature to provide these subjective rankings? The scores seem unrelated to the focal results of this paper. If the authors can provide some clearer justification for including these rankings (ideally with some measure of inter-rater reliability), I would also want to see a much more detailed description of what each rating means and how it is determined. What is the practical difference between a 2 and a 3 when it comes to the abstract and title? The sampling? The ethics? What meaning should a reader take away from these ratings?

A: Thank you for your comment. The relevant content was revised and elaborated if necessary.

In the Method section, “Quality assessment” and “Data extraction and analysis” were separated for clarification. In the Result section, “Quality of included studies” was also revised to clearly state the results of the study quality assessment. Please refer to below,

2.2.3. Quality assessment

The study quality of the included articles was assessed according to a reported structured questionnaire [48]. This tool evaluates multiple facets of a study from nine aspects. The total score of the scale ranges from 9 to 36, while the mean score is 22.5. The criteria (good, fair, poor, and very poor) with detailed descriptions of the ratings are shown in the Supplemental Material. Two authors independently completed the study quality assessment. The structured frame, i.e., study design, samples, and key measurement(s), were used to assess the eligible studies, which is shown in Supplemental Material Table S1.

2.2.4. Data extraction and analysis

Two authors independently extracted the data. Any discrepancies in the results were resolved through consultation with a third independent reviewer. A consensus was reached via discussions. The authors of this review firstly assigned keywords to identify the design and outcomes of the pedagogical programs using digital technology for assisting the integrated STEM education in children at early years. Secondly, any relevant outcomes were extracted from both quantitative analyses (e.g., intragroup or intergroup difference, etc.) and qualitative interviews (e.g., extracted themes). Thirdly, data extraction from the included studies was achieved by a thematic approach via a standardized table (see Table 2). This table includes study design, place of study conducted, basic information of participants, technology involved in the STEM education, the duration of the STEM program, and role(s) of technology played in young children’s learning and sensemaking (according to the specific definition/criteria of each category or step published in literature).

“3.2. Quality of included studies

The total score of study quality ranged between 18-35 for the included studies, whereas 82.4% of studies (14/17) had a total score higher than the mean score at 22.5 based on the questionnaire of Hawker et al. [48]. This indicates the relatively high quality of data provided by the reviewed studies. Studies with obvious shortcomings in sampling, ethics and bias, and transferability received lower scores based on the criteria. This was frequently related to a small sample size and/or a non-experimental design.”

Furthermore, the Table of study quality assessment was moved to Supplemental Material as Table S1, whereas the criteria with detailed descriptions of the ratings are following the table. This information is listed below,

* Criteria of ratings followed the statements listed as below,

  1. Abstract and title: Did they provide a clear description of the study?

Good (4) - Structured abstract with full information and clear title.

Fair (3) - Abstract with most of the information.
Poor (2) - Inadequate abstract.

Very Poor (1) - No abstract.

  1. Introduction and aims: Was there a good background and clear statement of the research aims?

Good (4) - Full but concise background to discussion/study containing up-to-date literature review and highlighting gaps in knowledge. A clear statement of aim AND objectives, including research questions.

Fair (3) - Some background and literature review. Research questions outlined.

Poor (2) - Some background but no aim/objectives/questions, OR Aims/objectives but inadequate background.

Very Poor (1) - No mention of aims/objectives. No background or literature review.

  1. Method and data: Is the method appropriate and clearly explained?

Good (4) - The method is appropriate and described clearly (e.g., questionnaires included)—clear details of the data collection and recording.

Fair (3) - The method is appropriate, but the description could be better. Data described.

Poor (2) - Questionable whether the method is appropriate. The method is described inadequately—little description of data.

Very Poor (1) - No mention of method, AND/OR Method inappropriate, AND/OR No details of data.

  1. Sampling: Was the sampling strategy appropriate to address the aims?

Good (4) - Details (age/gender/race/context) of who was studied and how they were recruited. Why was this group targeted? The sample size was justified for the study. Response rates are shown and explained.

Fair (3) - Sample size justified. Most information given, but some still needs to be included.

Poor (2) - Sampling mentioned but few descriptive details.

Very Poor (1) - No details of the sample.

  1. Data analysis: Was the description of the data analysis sufficiently rigorous?

Good (4) - Clear description of how the analysis was done. Qualitative studies: Description of how themes derived/respondent validation or triangulation. Quantitative studies: Reasons for tests selected hypothesis driven/numbers add up/statistical significance discussed.

Fair (3) - Qualitative: Descriptive discussion of analysis. Quantitative.

Poor (2) - Minimal details about analysis.

Very Poor (1) - No discussion of analysis.

  1. Ethics and bias: Have ethical issues been addressed, and what has necessary ethical approval gained? Has the relationship between researchers and participants been adequately considered?

Good (4) - Ethics: Where necessary confidentiality, sensitivity, and consent issues were addressed. Bias: The researcher was reflexive and/or aware of their own bias.

Fair (3) - Lip service was paid to the above (i.e., these issues were acknowledged).

Poor (2) - Brief mention of issues.

Very Poor (1) - No mention of issues.

  1. Results: Is there a clear statement of the findings?

Good (4) - Findings explicit, easy to understand, and in a logical progression. Tables, if present, are explained in the text. Results relate directly to aims. Sufficient data are presented to support findings.

Fair (3) - Findings mentioned, but more explanation could be given. Data presented relate directly to results.

Poor (2) - Findings presented haphazardly, not explained, and need to progress logically from results.

Very Poor (1) - Findings not mentioned or do not relate to aims.

  1. Transferability or generalizability: Are the findings of this study transferable (generalizable) to a wider population?

Good (4) - The context and setting of the study are described sufficiently to allow comparison with other contexts and settings, plus a high score in Question (d) (sampling).

Fair (3) - Some context and setting are described, but more is needed to replicate or compare the study with others, PLUS a fair score or higher in Question (d).

Poor (2) - Minimal description of context/setting.

Very Poor (1) - No description of context/setting.

  1. Implications and usefulness: How important are these findings to policy and practice?

Good (4) - Contributes something new and/or different in terms of understanding/insight or perspective. Suggests ideas for further research. Suggests implications for policy and/or practice.

Fair (3) - Two of the above (state what is missing in comments).

Poor (2) - Only one of the above.

Very Poor (1) - None of the above.

Retrieved from: Appendix D (p. 1296-1297) of Hawker, S., Payne, S., Kerr, C., Hardey, M., & Powell, J. (2002). Appraising the evidence: Reviewing disparate data systematically. Qualitative Health Research, 12(9), 1284-1299. doi:10.1177/1049732302238251

There are also some organizational issues with this paper, most notably - the text that is currently presented in the limitations section covers limitations of the studies that are reviewed. If comparing these limitations is a focus of the review, this analysis belongs in the results section. The limitations section should focus on the current study's limitations (the systematic review). 

A: Thank you for your comment. We agree with the reviewer. Thus, we moved the limitations acknowledged in the reviewed studies to the Result section, i.e., “3.7. Acknowledged limitations in the included studies & implications for research design”. The limitation in the Discussion section focused on the limitations of the current systematic review. Please see below,

“4.4. Limitations of this review

Aside from the fact that some of the included studies possessed limitations that may have affected the findings of the present review, this work also has limitations. Articles were retrieved from seven electronic databases, and only those including the specific terms mentioned in the title or abstract were reviewed for further analysis. Studies that were not in English, published in conference abstracts, government reports, textbooks, or unpublished dissertations were not included. The dataset includes the years 2020 and 2021 due to the scope of the study. The subsequent research agenda will include the COVID-19 and the subsequent post-pandemic period to understand the changes. Also, cutting-edge technologies (e.g. AI) may be included in the keywords to understand the new trends of digital technologies in early STEM education.”

Table 2 and Table 3 both include all 22 studies, but they are numbered differently, which makes it very difficult to look at results across the tables.

A: Thank you for your comment. Table 2 and Table 3 were revised, whereas the current 22 studies were numbered the same.

Section 2.2 Selection Criteria should really come before section 2.1 Search Strategy. Without knowing the selection criteria and the inclusion/exclusion goals, it is very difficult to make sense of the search strategy.

A: Thank you for your comment. Selection Criteria was moved to the first place in the Method section.

We need MUCH more detail about the search strategy, especially when large swaths of the original sample are removed. For example, at the stage of title/abstract review, what were the authors looking for that would help them determine that an article should be included/excluded? For the eight articles that were added by hand, why were they added? What did the authors do differently that they wouldn't have come up with from the original search strategy?

A: Thank you for your comment. We added a final step included in the search strategy (Table 1) and a detailed screening process in the Method section (2.2.2. Screening Process, please see below).

“2.2.2. Screening process

Two authors conducted the article screening. During the process, a consensus by comparison or discussion was made between the authors. After the exclusion by the automation tool of databases, titles/abstracts of 1,579 articles were screened. The titles and abstracts of the articles were screened to determine their relevance to the focus of this review. After the title/abstract screening, the remaining 240 articles were read in full to determine if they met the inclusion criteria. Another eight research papers were included by checking through the reference lists of the relevant published literature. Finally, 22 relevant articles were identified.”

Comments on the Quality of English Language

The writing of this manuscript needs significant revision. Nearly every sentence (with the exception of the results section, which is significantly better overall, though still includes errors) includes some grammatical errors, awkward word choice, and/or poor sentence structure. The errors do not render the paper incomprehensible but rather make the writing quite clunky and difficult to follow.

A: Thank you for your comment. The current manuscript was reviewed and edited by a native English speaker.

Round 2

Reviewer 3 Report

Comments and Suggestions for Authors

I appreciate the care and attention that went into this revision. Many of my concerns have been addressed, however, I still have a major concern with the scope of the review being limited to studies that include the word "STEM." The current limitations section does not go far enough in explaining this limitation. The authors should make it abundantly clear that this review only covers the tip of the iceberg by including articles that explicitly mention STEM learning.

Comments on the Quality of English Language

Much improvement in this revision, however further English-lanugage editing is necessary.

Author Response

Response to reviewer’ comments

RE: education-2759536

Title: Roles and effect of digital technology in young children’s STEM learning: A scoping review of empirical studies

Reviewer 3

I appreciate the care and attention that went into this revision. Many of my concerns have been addressed, however, I still have a major concern with the scope of the review being limited to studies that include the word "STEM." The current limitations section does not go far enough in explaining this limitation. The authors should make it abundantly clear that this review only covers the tip of the iceberg by including articles that explicitly mention STEM learning.

A: Thank you so much for your comment.

To address this comment, firstly, we changed the title of this study to “scoping review”, please refer to the title, title page, and the relevant description marked in the main text.

Secondly, we addressed and extended this limitation in the Discussion section (4.4. Limitations of this review), please refer to the main text (page 10, lines 593-601) or below,

“To emphasize and focus on the multi-/inter-disciplinary nature of STEM learning, this scoping review initiated this topic with a relatively narrow search term (i.e., “STEM”). This may only cover the tip of the iceberg by including articles that explicitly mention “STEM” learning. Similarly, articles were retrieved from seven electronic databases, and only those including the specific terms mentioned in the title or abstract were reviewed for further analysis. As a result, future reviews may extend the search terms from including “STEM” only to all single disciplines and investigate and discuss in a broader scope.”

Much improvement in this revision, however further English-language editing is necessary.

A: Thank you so much for your comment. The current version was edited and revised by another native speaker who provided a professional English editing service.